# Maturity Offset, Anthropometric Characteristics and Vertical Force–Velocity Profile in Youth Basketball Players

**DOI:** 10.3390/jfmk8040160

**Published:** 2023-11-17

**Authors:** Pablo Jiménez-Daza, Luis Teba del Pino, Julio Calleja-Gonzalez, Eduardo Sáez de Villarreal

**Affiliations:** 1Physical Performance Sports Research Center (PPSRC), Universidad Pablo Olavide Sevilla, 41013 Sevilla, Spain; pjimenez1183@gmail.com (P.J.-D.); coach.luisteba@gmail.com (L.T.d.P.); 2Physical Education and Sport Department, Faculty of Education and Sport, University of the Basque Country (UPV/EHU), 01007 Vitoria, Spain; julio.calleja.gonzalez@gmail.com

**Keywords:** muscle strength, team sports, athletic performance, youth sports, human physical conditioning

## Abstract

This work aimed to analyze the relationships between maturity offset, anthropometric variables and the vertical force–velocity profile in youth (12–18 years old) male basketball players. The vertical force–velocity profile was measured in 49 basketball players, grouped in competitive-age categories, i.e., under 14, 16 and 18 years of age (U-14, U-16 and U-18, respectively). A bivariate correlational analysis was carried out between maturity offset, anthropometric variables (height, body mass, % fat, muscle mass, bone mass and body mass index (BMI)) and vertical force–velocity profile (theoretical maximal force [F0], theoretical maximal velocity [V0], theoretical maximal power [Pmax], force–velocity imbalance [Fvimb] and force–velocity profile orientation). The results showed significant correlations (*p* < 0.05) between Fvimb and maturity offset at early ages of training (12–15 years). The anthropometric profile was correlated (*p* < 0.05) with F0 in U-14, V0 in U-16, and Pmax in U-18 basketball players. The current findings suggest a relationship between the vertical force–velocity imbalance and maturity offset and the main vertical force–velocity profile variables. The vertical force–velocity profile is hypothesized as a useful index to correct vertical force–velocity deficits according to the maturity offset of male basketball players.

## 1. Introduction

Basketball performance is determined by physical, physiological and morphological variables [1,2]. These variables are reflected in the anthropometric characteristics of athletes (height, body mass and wingspan), which are of great relevance both in elite sports performance [3,4] and at high-school ages [5], as well as in the prediction of basketball talent [6]. Current trends have investigated and developed the so-called maturity offset (MO), a formula that has been created and nourished by the idea of an anthropometric profile, which helps to know at what point of maturation, understood as the distance between the current growth velocity and its maximum possible (peak height velocity; PHV), is an athlete. The idea of maturity offset arises as a response to a present problem in the world of training sports: the wide anthropometric differences (especially in height and body mass) between players of the same chronological age, which is a very limited tool for evaluating and establishing the maturational and growth profile of athletes [7] since, in athletes, especially at adolescent age, the inter-subject maturational variability is too large to establish general standards based on chronological age [8]. Based on the maturity offset, it is possible to individualize the work to exploit the potential performance of basketball players. Success in basketball is largely related to the physical and anthropometric characteristics of players [9], both individually and collectively. So, controlling the maturational age will facilitate greater assimilation of concepts and lead to greater individualization of training, avoiding early specializations and work in vain [10]. In this sense, the maturity offset, being an individualizing tool, helps us to meet all the above requirements. Several studies highlight the relationship between the maturity offset of players in training and their jumping and sprinting ability [11,12], which are clear variables to be considered in basketball. In addition, the correct development of these capacities is determinant in the force–velocity relationship [13].

On the other hand, the correct development of basic physical abilities and those specific to basketball are also key to the direct and future performance of athletes [14]. The ability to jump or sprint is basic for performance in general [15] and for basketball, in particular [16], since it will allow players to jump more, accelerate more and decelerate with a better technique (reducing the risk of injury) or sprint [17,18]. In this line, the work and development of maximal power and maximal force in the training stage are recommended since they seem to be two determining factors in talented players [19]. Therefore, the evaluation and monitoring of adaptations and variations at the neuromuscular level, induced by specific training (i.e., variables that predominate in a sport and on which performance depends), becomes important [20]. The vertical force–velocity profile (FVP) meets the requirements of specificity and individualization since it visualizes, by vertical jump performance, the strong relationship between the force applied by a subject and the velocity at which the subject applies that force per unit time (RFD) [15,21,22,23,24], thus establishing individualized imbalances, either force or velocity [25], that will guide the intervention program until the optimization of the vertical force–velocity profile is achieved. The vertical force–velocity profile has been used and contrasted in the high performance of collective sports: basketball [26], soccer [27], handball [24], rugby [28] and water polo [29], among others. It has even been tested in young soccer players in training [30], but not in basketball players at maturing age. However, it is possible that the variations present in the variables associated with vertical force–velocity profiles, including maximum theoretical force (F0), maximum theoretical velocity (V0), maximum theoretical power (Pmax), force–velocity profile imbalance (Fvimb), and the type of deficit of the force–velocity profiles, are more related to maturity offset (MO) and anthropometric characteristics than to chronological age [30].

Therefore, given the relevance of anthropometry, maturation and performance of basketball players, the present study aims to (i) analyze and develop the possible practical relationships between the index obtained from the maturity offset and the imbalance presented in the force–velocity profile in male young basketball players (12–18 years old), and (ii) to find out possible correlations of practical utility between the vertical force–velocity profile and the anthropometric variables included in the study. Consequently, despite the lack of previous literature on youth basketball players, the combination of both methodologies may offer an interesting research door, i.e., it is reasonable to assume that there will be significant correlations between maturity offset and force–velocity profile.

## 2. Methods

### 2.1. The Experimental Approach to the Problem

This study presented a descriptive and correlational design, in which data corresponding to the MO and FVP of 49 male young basketball players were collected.

### 2.2. Participants

The research sample consisted of 49 male youth basketball players (from the same Basketball Club) who were differentiated by category (U-14; U-16; U-18) (Table 1). All the players in the sample played in the highest division of their category. The intervention protocol and the methodology proposed for the present investigation followed the regulations established in the latest version of the Declaration of Helsinki and the ethics committee of the responsible department approved the study protocol. Finally, all the subjects participating in this study received the corresponding explanations regarding the objectives, procedures and obligations involved in the research and signed the corresponding informed consent form.

### 2.3. Procedures

First, we proceeded to measure the vertical force–velocity profile of the participating teams, for which two and a half months were necessary. The days invested in each team ranged between 2 and 4, depending on the presence and accessibility of the players, and the facilities and equipment available during the competitive period. Measurements were taken during the two hours per week that each team devoted to physical preparation so that the time invested in each team ranged between 2 and 4 h. The data recorded were subject, weights with which used to perform the test, force–velocity profile determination coefficient, theoretical maximum power, force–velocity imbalance and type of the deficit, theoretical maximum force and theoretical maximum velocity. In the sample participating in this study, a large majority of force imbalances (high or low) were found, along with a large number of optimal and balanced force–velocity profiles and a small percentage of subjects recorded velocity deficits. It should be noted that the force–velocity profile measurements complied with the recommendations of Jiménez-Reyes et al. (2014) [23] and were taken at least two days after a match in order to facilitate the correct recovery of the athletes and minimize the influence of residual fatigue.

Secondly, the measurement of MO and the collection of anthropometric data required one day of measuring per team. The variables recorded included height, sitting height, leg length, age, body mass (variables included in the MO), body fat percentage, muscle mass, bone mass, BMI, the fat percentage in each leg, and muscle mass in each leg. The tests were performed at the club’s stadium, with which the players were already familiar, as well as with the club’s materials (bars, weights), which were the same that they had used throughout the season. The players attended the testing sessions with the appropriate sports equipment to carry out the different evaluations (footwear, clothing and equipment). Adequate rest was recommended, with no strenuous activity in the 24 h before the test, as well as proper intake and hydration to avoid biased data that could lead to erroneous results.

For the measurement of the force–velocity profile, it started with a standardized 10–15 min warm-up for all the subjects who participated in this study, which consisted of joint mobility, dynamic and ballistic stretching, isometric activation and test approach exercises. The measurement of the force–velocity profile of each subject was carried out according to Samozino’s method [31], for which the MyJump2 app (iPad IOS mini 2020 Slow Motion) was used, as already been used in previous studies [32,33].

For the construction of the force–velocity profile, each subject was subjected to a test of between 3 and 5 incremental loads, which presented high validity [15]. The first jump was performed only with their own body weight plus pike (0.5 kg) and, subsequently, the weight was increased jump after jump. Since all subjects previously performed CMJ jumps with and without weights, the 5 weights were individually adapted to each subject. The optimal angle at which to start the push-off phase was chosen by the players, which was always the same for each jump [33]. The countermovement jump was used since it is much more similar to the demands and dynamic movements of basketball. As of today, its use is widely validated for the measurement of force–velocity profile [23,34]. Finally, when the corresponding load allowed a jump close to 10 cm in height, the test was terminated [34].

For the maturity offset assessment, the equations of Mirwald et al. (2002) [7] were used, differentiated according to sex:MO (Male) = 9.236 + 0.002708 × (leg length × sitting height) − 0.001663 × (age × leg length) + 0.007216x (age × sitting height) + 0.02292 × (height × body mass)(1)

The subjects were asked to be adequately hydrated and not to have ingested food in the 2 h before the measurements. For the height measurement, each subject was placed barefoot in front of a tape measure in a vertical position, perfectly positioned from the ground. For the sitting height measurement, a chair without a backrest was placed so that the subject could sit as upright as possible. The subject’s feet had to be suspended in the air or, if this did not occur, the angle formed by the knees had to be greater than 90 degrees. The sitting height was calculated as the distance from the height of the bench to the height reached by the subject. The leg length for maturity offset was calculated as the difference between the total height and the sitting height. All measurements were adjusted to the centimeter.

All other anthropometric variables were collected using a Tanita-BC-601 (Tokyo, Japan). The subjects were asked to stand upright and to place their feet and hands correctly on the eight electrodes; the elbows had to be fully extended. The data were as follows: body mass, muscle mass, bone mass, fat index, BMI, the muscle mass of the right leg, the muscle mass of the left leg, the fat percentage of the right leg and the fat percentage of the left leg.

### 2.4. Statistical Analyses

Statistical analysis was performed using SPSS version 25 for Windows 10 (SPSS, Inc., Chicago, IL, USA). All data were subjected to descriptive statistical analysis of correlations. The Shapiro–Wilk test was carried out to check the normality of the sample distribution. Correlations were determined using Pearson’s bivariate correlation coefficient (R). Based on statistical theory, any relationship with coefficient 1 will be considered a perfect positive correlation; any relationship with coefficient −1 will be considered a perfect negative correlation; and any relationship with coefficient 0 will assume that there is no linear relationship between the variables. When |R| is <0.4, the relationship will be low; if |R| > 0.4, the correlation will be of medium degree; if |R| > 0.7, the correlation will be interpreted as good/high. For all variables recorded, the results were adjusted to the second decimal place.

## 3. Results

The analysis was performed by categories in order to understand in which age ranges the Fvimb-MO relationship was more critical (Table 2, Table 3 and Table 4).

In the U-14 category, correlations (*p* < 0.05) were observed between force–velocity profile imbalance and maturity offset (R = −0.458; *p* = 0.05). An inverse relationship was noted between height and Fvimb (R = −0.370; *p* = 0.05). Highly significant correlations were observed between MO and FVP_Def (R = 0.471; *p* = 0.01) and between MO and F0 (R = 0.619; *p* = 0.01). A strong relationship between F0 and maturity offset was noted (R = 0.619; *p* = 0.01). There were also strong correlations between F0 and height (R = 0.476; *p* = 0.01), F0 and body mass (R = 0.400; *p* = 0.05), F0 and total muscle mass (R = 0.490; *p* = 0.01), F0 and bone mass (R = 0.478; *p* = 0.01), and finally, F0 and muscle mass of the right and left legs (R = 0.421, *p* = 0.05; and R = 0.405, *p* = 0.05, respectively).

In the U-16 category, a highly significant correlation was observed between maturity offset and force–velocity profile imbalance (R = 0.542; *p* = 0.01). It was noted a direct correlation between force–velocity profile imbalance and height (R = 0.772; *p* = 0.01). Strong correlations were noted between Fvimb and total muscle mass (R = 0.502; *p* = 0.01) and also between Fvimb and bone mass (R = 0.509; *p* = 0.01). Correlations were observed between MO and FVP_Def (R = −0.550; *p* = 0.01), MO and Pmax (R = 0.585; *p* = 0.01), and MO and V0 (R = 0.615; *p* = 0.01). It was noted strong relationships between height and the following variables: Pmax (R = 0.532; *p* = 0.01), V0 (R = 0.670; *p* = 0.01), F0 (R = −0.549; *p* = 0.01) and FVP_Def (R = −0.515; *p* = 0.01). Total muscle mass presented correlations with V0 (R = 0.595; *p* = 0.01), Pmax (R = 0.528; *p* = 0.01) and FVP_Def (R = −0.440; *p* = 0.05). There were observed correlations between muscle mass of the right leg and the following variables: V0 (R = 0.547; *p* = 0.01), F0 (R = −0.396; *p* = 0.05), Pmax (R = 0.480; *p* = 0.05) and FVP_Def (R = −0.454; *p* = 0.05). It was noted important relationships between the muscle mass of the left leg and the following variables: V0 (R = 0.618; *p* = 0.01), F0 (R = −0.402; *p* = 0.05) and Pmax (R = 0.528; *p* = 0.01).

Finally, in the U-18 category, there was no relationship between force–velocity imbalance and maturity offset (*p* = 0.274 > 0.05), although maturity offset presented a significant correlation with theoretical maximum power ((R = 0.408; *p* = 0.05). There were correlations between Pmax and height (R = 0.388; *p* = 0.05), Pmax and body mass (R = 0.510; *p* = 0.01), Pmax and total muscle mass (R = 0.483; *p* = 0.01), Pmax and bone mass (R = 0.545; *p* = 0.01), Pmax and BMI (R = 0.569; *p* = 0.01), Pmax and muscle mass of the right leg (R= 0.502; *p* = 0.01), and Pmax and muscle mass of the left leg (R = 0.493; *p* = 0.01). The V0 correlated with BMI (R = 0.487; *p* = 0.01), body mass (R = 0.413; *p* = 0.05), muscle mass (R = 0.422; *p* = 0.05), right leg muscle mass (R = 0.419; *p* = 0.05) and left leg muscle mass (R = 0.428; *p* = 0.05).

## 4. Discussion

The main results revealed a correlation between maturity offset and vertical Fvimb in the U-14 and U-16 basketball age groups, although not in the U-18 age category. Moreover, a high correlation was noted between F0 and height, body mass, muscle mass and bone mass in U-14 players. In U-16 players, V0 and Fvimb were highly correlated with anthropometric variables. In the U-18 category, Pmax was correlated to anthropometric variables. The current novel findings and their implications are discussed in the following paragraphs.

The data revealed greater force–velocity deficits in participants whose MO was farther away from the peak height velocity (i.e., maturity offset away from “0”). The main growth windows of peak height (i.e., peak height velocity) may occur at 12 and 15 years of age (29) and, therefore, in the transition from the last year of the U-12 category to the U-14 category, which does not appear reflected in the present study since almost all subjects born in that year presented maturity offset far from 0 (close to −2). However, there is a growth window in the U-16 category (14–15 years) since 65% of the subject’s presented maturity offset between −0.9 and 0.9, i.e., close to 0, and no athlete reached two units of difference (maximum value: 1.6). Therefore, FVP enhancement and assessment may be better suited in U-16 players [35].

Due to the MO-Fvimb relationship in the U-14 and U-16 categories, the correlation went from inverse to direct with a greater age category. This may be explained as most U-14 players had negative maturity offset values (i.e., <0), while most U-16 players had positive maturity offset values. Therefore, an inverse MO-Fvimb relationship may be plausible among U-14 and U-16 players since lower maturity offset values correlate with greater Fvimb values. This progression may partially explain the lack of relationship between Fvimb and MO among U-18 players, as is expected a similar level of biological maturation at this age stage [35]. Therefore, given that the average maturity offset is clearly higher than that of the other two categories and given that it is the only one in which the totality of the subjects’ present positive indices (i.e., they have already exceeded the PHV), the relationship between force–velocity imbalance and maturity offset would be meaningless.

Finally, a novel finding not contemplated in the initial hypotheses is the discovery of the so-called “determinant variable”, which could be defined as “the variable of the vertical force-velocity profile that presents the greatest number of significant correlations with the anthropometric data measured and, therefore, is the one that offers us the greatest combined information”. As mentioned at the beginning of this section, depending on the category, the determinant variable changes. First, it is observed that at early ages (12–13 years), theoretical maximum force correlates directly and significantly with all anthropometric variables (Table 2), except for %fat (total and both legs) and BMI. Furthermore, the high intensity of the correlation with maturity offset shows us that, in the U-14 category, the taller, heavier and stronger (understanding strength as a greater amount of muscle mass and bone mass) a subject is, the greater his F0 will be, which leads to a lower imbalance (lower force deficit). In reference to the literature, it is normal that, in subjects with little experience in resistance training, the F0 component takes precedence over the V0 component since the force–velocity spectrum is a complex index based on the efficient use of force, which translates into a higher Pmax and a higher RFD [18,36,37]. In such young and inexperienced male players, the ability to generate high levels of force per unit of time goes through a correct understanding of the technique of the different strength exercises, for which exercises with controlled, non-explosive tempos are required (which will generate a lower RFD). In this line, height, body mass, muscle mass and bone mass at these ages are clear symbols of maturational age [2,4,9], so it is understandable that the F0 of a player depends directly on his maturational status. Likewise, greater heights will, in many cases, imply a longer lower body, i.e., greater height of push-off (Hpo), so athletes will have the capacity to reach higher levels of force.

V0 and Fvimb take center stage in subjects aged 14 and 15 years. In this case, V0 follows the same correlational line as F0 in U-14 players: it presents highly significant relationships with the great majority of anthropometric variables, except for BMI and %fat of both legs (Table 3). Once again, the relevance of maturity, height and mass (total body, muscle and bone) with V0 stands out. At a general level, the importance of V0 is explained by the growth and maturation of an athlete (greater maturity offset leads to greater V0), who is in a middle stage with a certain amount of experience in resistance training. Thanks to this, the athlete will be able to take more advantage of the velocity component of the force–velocity curve, i.e., will generate more force per unit of time [20,25]. This causes a more efficient use of force, which gains even more importance in a sport like basketball where ballistic performance and the ability to perform explosive actions are of vital importance [2,4,14,19,35,38,39,40]. On the other hand, force–velocity imbalance takes on an even more protagonist and complex role at these ages since male and female players with greater height, muscle mass and bone mass will tend to have higher force imbalances. Following this line of argument, it is normal that subjects with longer Hpo will continue to prioritize the force component over the velocity component. In addition, at a practical level, taller players at mature ages tend to have greater difficulty in assimilating a correct technique in exercises with external loads (i.e., force–velocity profile), so, in these cases, the speed of execution will depend to a greater extent on the level of control of the body itself and the extra load.

Finally, in U-18 players, Pmax was highly correlated with body mass, muscle mass and bone mass. Therefore, the influence of body mass [1] and strength capacity [2,18] on performance during the adolescent stages is confirmed. The percentage of body fat is negatively correlated with Pmax [41]. Higher Pmax values may be related to a more balanced force–velocity profile and, therefore, an optimized RFD [29,36,42], with the V0 component being the focus of work in most cases [43,44]. Some of the most relevant studies of the force–velocity profile has already related Pmax to ballistic performance [15,36], so it is understandable that in the final stage of high-school training, where players who stand out are practically treated as professionals, the determinant variable is Pmax [2]. However, the sample of the present study shows that the Pmax of all of these subjects could be even higher since a large number of male players present force–velocity profiles with high/low deficiencies in strength, which directly influence the production of strength per unit of time and Pmax. Therefore, an intervention from the early categories focusing on RFD and Pmax work could help certain players to develop better sports performance during the adolescent stages [2,18,21]. All this always takes into account the maturity offset [7] or biological age of athletes [37] and their individualized force–velocity imbalance [15,25,42]).

## 5. Conclusions

In conclusion, in the U-14 and U-16 categories, a MO-FVP relationship was observed. Variables such as body mass, height, muscle mass (total, and right and left legs), bone mass, F0 (especially in U-14), V0 (especially in U-16) and Pmax (especially in U-18) may be considered in the long-term monitoring–training process of youth basketball players.

## 6. Limitations and Strengths

The main limitation of the present study was the difficulty in measuring the vertical force–velocity profile of the players in a shorter space of time. This was sometimes due to the unavailability of the space necessary to carry out the midfields, and on other occasions, due to the non-presence of all players on the planned day to take the measurements. This was solved by increasing the number of days available to the players who participated the study to carry out the measurements.

Due to the early specialization of basketball players, the need to establish strategies that allow optimizing the learning and performance of players based on their maturing age increases. The results of the present study could help coaches and physical trainers of the U-14, U-16 and U-18 categories of men’s basketball to design individual strength training for young basketball players considering MO, thereby optimizing their present and future performance.

## 7. Practical Applications

The current findings suggest that youth players (U-14 and U-16) with greater height, body mass, bone mass and muscle mass should develop RFD and Pmax to reduce force–velocity imbalance in future categories (U-18 and professional leagues) and, thus, optimize their present and future performance. Therefore, the use of maturity offset as a complementary tool to the vertical force–velocity profile can offer a way of early intervention for young basketball players. The present study may help practitioners to better assess MO in youth male U-14, U-16 and U-18 competitive basketball age categories, and its relationship with their somatic growth and the vertical force–velocity profile (F0; V0; Pmax; and Fvimb).

## Figures and Tables

**Table 1 jfmk-08-00160-t001:** Participants’ descriptive characteristics according to competitive-age category.

	U-18 (16–17 Years) *	U-16 (16–17 Years) *	U-14 (16–17 Years) *
Subject (n)	16	16	17
Anthropometric data (mean ± SD)			
BW (kg)	92.72 ± 12.09	72.10 ± 10.20	55.15 ± 10.52
Height (cm)	193.18 ± 6.06	186.19 ± 7.61	171.71 ± 11.03
Sitting height (cm)	97.27 ± 2.28	94.31 ± 3.09	85.88 ± 5.38
Leg length (cm)	95.91 ± 4.50	91.88 ± 5.21	85.82 ± 7.22
Age (years)	17 ± 0.70	15 ± 0.63	13 ± 0.61
BMI (Kg/m^2^)	24.77 ± 2.37	20.71 ± 1.91	18.51 ± 1.69
%fat	15.90 ± 5.42	8.71 ± 3.96	8.59 ± 3.33
%fat_r	14.21 ± 4.24	8.18 ± 5.43	7.76 ± 3.85
%fat_l	14.43 ± 4.28	7.99 ± 5.63	8.05 ± 3.62
Bm (kg)	3.85 ± 0.29	3.28 ± 0.39	2.45 ± 0.46
Mm (kg)	74.07 ± 6.71	62.38 ± 8.04	47.86 ± 9.43
Mm_r (kg)	12.75 ± 1.57	10.54 ± 1.30	8.68 ± 1.78
Mm_l (kg)	12.55 ± 1.62	10.74 ± 1.37	8.33 ± 1.73
MO (years)	2.48 ± 0.44	1.04 ± 0.50	−4.44 ± 4.22
Determinant variable (mean ± SD)			
F0 (N/kg)	27.11 ± 3.35	28.67 ± 4.43	28.38 ± 6.84
V0 (m/s)	4.29 ± 0.91	3.90 ± 0.96	3.47 ± 1.27
Fv_imb (%)	52.26 ± 14.67	43.05 ± 18.12	44.59 ± 20.01
Pmax (W/kg)	28.58 ± 4.46	27.43 ± 5.37	23.24 ± 5.57

* Chronological age of each category. BW: Body Weight; Fv_imb: FVP Imbalance; MO: Maturity Offset; %fat: Fat Index; Mm: Muscle Mass; Bm: Bone Mass; BMI: Body Mass Index; Mm_r: Right leg muscle mass; Mm_l: Left leg muscle mass; %fat_r: Right leg fat index; %fat_l: Left leg fat index; F0: Theoretical Maximum Force; V0: Theoretical Maximum Velocity; Pmax: Theoretical Maximum Power.

**Table 2 jfmk-08-00160-t002:** Inter-correlation matrix between MO, FVP variables and anthropometric variables in the U-14 category.

	Fvimb	MO	Height	Body Mass	%fat	Mm	Bm	BMI	Mm_r	Mm_l	%fat_r	%fat_l	FVP_Def	F0	V0	Pmax
Fv_imb	1															
MO	−0.458 *	1														
Height	−0.370 *	**0.810** **	1													
Weight	−0.295	**0.548** **	**0.766** **	1												
%fat	−0.070	−0.308	−0.190	0.295	1											
Mm	−0.288	**0.691** **	**0.870** **	**0.938** **	−0.048	1										
Bm	−0.278	**0.678** **	**0.867** **	**0.930** **	−0.063	**0.997** **	1									
BMI	−0.150	0.097	0.255	**0.799** **	**0.692** **	**0.587** **	**0.576** **	1								
Mm_r	−0.341	**0.711** **	**0.731** **	**0.474** **	−0.362 *	**0.617** **	**0.615** **	0.067	1							
Mm_l	−0.341	**0.694** **	**0.722** **	**0.486** **	−0.332	**0.617** **	**0.613** **	0.094	**0.996** **	1						
%fat_r	−0.133	−0.311	−0.295	0.185	**0.870** **	−0.123	−0.141	**0.609** **	**−0.489** **	−0.439 *	1					
%fat_l	−0.154	−0.273	−0.264	0.226	**0.898** **	−0.089	−0.107	**0.645** **	**−0.470** **	−0.436 *	**0.975** **	1				
FVP_Def	−0.367 *	**0.471** **	0.350	0.284	−0.018	0.290	0.269	0.121	0.433 *	0.433 *	−0.056	−0.014	1			
F0	−0.383 *	**0.619** **	**0.476** **	0.400 *	−0.188	**0.490** **	**0.478** **	0.170	0.421 *	0.405 *	−0.118	−0.096	**0.633** **	1		
V0	**0.799** **	−0.426 *	−0.283	−0.302	−0.169	−0.247	−0.238	−0.255	−0.261	−0.260	−0.266	−0.286	**−0.467** **	**−0.659** **	1	
Pmax	**0.776** **	−0.156	−0.012	−0.017	−0.242	0.072	0.073	−0.090	−0.057	−0.057	−0.333	−0.350	−0.233	−0.283	**0.876** **	1

Fv_imb: FVP Imbalance; MO: Maturity Offset; %fat: Fat Index; Mm: Muscle Mass; Bm: Bone Mass; BMI: Body Mass Index; Mm_r: Right leg muscle mass; Mm_l: Left leg muscle mass; %fat_r: Right leg fat index; %fat_l: Left leg fat index; PFV_Def: Type of deficit of FVP; F0: Theoretical Maximum Force; V0: Theoretical Maximum Velocity; Pmax: Theoretical Maximum Power. **. 0.01 significant correlation (bilateral). *. 0.05 significant correlation (bilateral). Category = U-14.

**Table 3 jfmk-08-00160-t003:** Inter-correlation matrix between MO, FVP variables and anthropometric variables in the U-16 category.

	Fvimb	MO	Height	Body Mass	%fat	Mm	Bm	BMI	Mm_r	Mm_l	%fat_r	%fat_l	FVP_Def	F0	V0	Pmax
Fv_imb	1															
MO	**0.542** **	1														
Height	**0.595** **	**0.772** **	1													
Weight	0.358	**0.510** **	**0.704** **	1												
%fat	−0.393 *	**−0.560** **	**−0.464 ***	0.079	1											
Mm	**0.502** **	**0.722** **	**0.858** **	**0.902 ****	−0.355	1										
Bm	**0.509** **	**0.719** **	**0.848** **	**0.907 ****	−0.342	**0.998 ****	1									
BMI	−0.149	−0.102	−0.062	**0.662 ****	**0.610 ****	0.357	0.376	1								
Mm_r	**0.509** **	**0.530** **	**0.725** **	**0.768 ****	−0.290	**0.850 ****	**0.856 ****	0.303	1							
Mm_l	0.496 *	**0.655** **	**0.829** **	**0.846 ****	−0.363	**0.952 ****	**0.951 ****	0.317	**0.774 ****	1						
%fat_r	−0.289	−0.314	−0.302	0.210	**0.830 ****	−0.165	−0.161	**0.637 ****	−0.318	−0.184	1					
%fat_l	−0.296	−0.364	−0.309	0.212	**0.868 ****	−0.179	−0.175	**0.644 ****	−0.247	−0.216	**0.955 ****	1				
FVP_Def	**−0.948** **	**−0.550** **	**−0.549** **	−0.263	0.458 *	−0.440 *	−0.447 *	0.238	−0.454 *	−0.386	0.348	0.370	1			
F0	**−0.854** **	−0.350	**−0.515** **	−0.313	0.243	−0.385	−0.403 *	0.121	−0.396 *	−0.402 *	0.262	0.275	**0.798 ****	1		
V0	**0.899** **	**0.615** **	**0.670** **	0.427 *	−0.433 *	**0.595 ****	**0.588 ****	−0.119	**0.547 ****	**0.618 ****	−0.251	−0.268	**−0.801 ****	**−0.628 ****	1	
Pmax	**0.593** **	**0.585** **	**0.532** **	0.334	−0.455 *	**0.528 ****	**0.509 ****	−0.097	0.480 *	**0.528 ****	−0.218	−0.239	**−0.525 ****	−0.155	**0.858 ****	1

Fv_imb: FVP Imbalance; MO: Maturity Offset; %fat: Fat Index; Mm: Muscle Mass; Bm: Bone Mass; BMI: Body Mass Index; Mm_r: Right leg muscle mass; Mm_l: Left leg muscle mass; %fat_r: Right leg fat index; %fat_l: Left leg fat index; PFV_Def: Type of deficit of FVP; F0: Theoretical Maximum Force; V0: Theoretical Maximum Velocity; Pmax: Theoretical Maximum Power. **. 0.01 significant correlation (bilateral). *. 0.05 significant correlation (bilateral). Category = U-16.

**Table 4 jfmk-08-00160-t004:** Inter-correlation matrix between MO, FVP variables and anthropometric variables in the U-18 category.

	Fvimb	MO	Height	Body Mass	%fat	Mm	Bm	BMI	Mm_r	Mm_l	%fat_r	%fat_l	FVP_Def	F0	V0	Pmax
Fv_imb	1															
MO	0.274	1														
Height	0.180	**0.810 ****	1													
Weight	0.283	**0.812 ****	**0.911 ****	1												
%fat	0.113	−0.087	−0.027	0.178	1											
Mm	0.239	**0.840 ****	**0.943 ****	**0.954 ****	−0.050											
Bm	0.302	**0.856 ****	**0.928 ****	**0.957 ****	−0.065	**0.981 ****	1									
BMI	0.381 *	**0.642 ****	**0.622 ****	**0.886 ****	0.401 *	**0.758 ****	**0.787 ****	1								
Mm_r	0.299	**0.850 ****	**0.886 ****	**0.967 ****	0.075	**0.952 ****	**0.953 ****	**0.844 ****	1							
Mm_l	0.314	**0.849 ****	**0.880 ****	**0.963 ****	0.096	**0.943 ****	**0.947 ****	**0.845 ****	**0.996 ****	1						
%fat_r	0.352	0.252	0.273	**0.480 ****	**0.701 ****	0.313	0.381 *	**0.658 ****	0.387 *	0.430 *	1					
%fat_l	0.385 *	0.281	0.285	**0.507 ****	**0.690 ****	0.366	0.420 *	**0.693 ****	0.438 *	0.458 *	**0.948 ****	1				
FVP_Def	**−0.693 ****	−0.087	0.059	−0.074	−0.095	−0.005	−0.066	−0.255	−0.090	−0.091	−0.213	−0.233	1			
F0	**−0.566 ****	−0.068	0.032	0.021	−0.142	0.114	0.028	−0.032	−0.004	−0.042	−0.247	−0.177	0.282	1		
V0	**0.930 ****	0.371	0.307	0.413 *	0.067	0.367	0.442 *	**0.487 ****	0.419 *	0.428 *	0.365	0.415 *	**−0.579 ****	**−0.506 ****	1	
Pmax	**0.814 ****	0.408 *	0.388 *	**0.510 ****	−0.020	**0.483 ****	**0.545 ****	**0.569 ****	**0.502 ****	**0.493 ****	0.293	0.371	**−0.561 ****	−0.170	**0.925 ****	1

Fv_imb: FVP Imbalance; MO: Maturity Offset; %fat: Fat Index; Mm: Muscle Mass; Bm: Bone Mass; BMI: Body Mass Index; Mm_r: Right leg muscle mass; Mm_l: Left leg muscle mass; %fat_r: Right leg fat index; %fat_l: Left leg fat index; PFV_Def: Type of deficit of FVP; F0: Theoretical Maximum Force; V0: Theoretical Maximum Velocity; Pmax: Theoretical Maximum Power. **. 0.01 significant correlation (bilateral). *. 0.05 significant correlation (bilateral). Category = U-18.

## Data Availability

To promote transparency of the data supporting the results reported in the article, the authors have established the data availability statement. The data associated with this article are not publicly available but are available from the corresponding author upon reasonable request.

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
