# Peer review of "Maturity Offset, Anthropometric Characteristics and Vertical Force–Velocity Profile in Youth Basketball Players"

_jfmk, 2023, doi:10.3390/jfmk8040160_

Round 1
Reviewer 1 Report
Comments and Suggestions for Authors The article provides interesting information and new approachesthat can be used for long-term monitoring of the training process
in young basketball players.
In the abstract, the term "determinant variable" is given in italics, which is explained later in the text on lines 151-155. It would be desirable to make the abstract more comprehensible.
In the "Subjects" section, it would be appropriate to add the performance level of the monitored teams of basketball players, were they elite category players?
In the “Statistical analysis” section, the normality of data distributionshould be explained, and the use of parametric statistical methods
should be justified.
The manuscript lacks tables with basic data, anthropometric data as well as F-V test values ​​and MO data for the three groups of players under observation.
The manuscript only presents the correlation matrix in the threeobserved groups, the key correlation relationships could be listed
and commented on in the text.
At the end of the discussion, the strengths and limitations of the presented study should be mentioned in a separate paragraph.
Author Response
Response to manuscript reviewers:
Maturity offset, anthropometric characteristics and vertical force-velocity profile in youth basketball players.
Dear Editor,
Next, we describe the changes made to the manuscript following the recommendations made by the reviewers:
Reviewer 1
Dear review,
First, thank you very much for taking your time to help us to improve our manuscript. We have considered all your suggestions and contributions. Finally, we have accepted and agreed to modify the text by applying recommendations number 1, 2, 3, 4 and 6.
- In the abstract, the term "determinant variable" is given in italics, which is explained later in the text on lines 151-155. It would be desirable to make the abstract more comprehensible.
Lines: 17.
Modification: “…and the main vertical force-velocity profile variables.”
- In the "Subjects" section, it would be appropriate to add the performance level of the monitored teams of basketball players, were they elite category players?
Lines: 93 - 94.
Modification: “All the players in the sample played in the highest division of their category”.
- In the “Statistical analysis” section, the normality of data distribution should be explained, and the use of parametric statistical methods should be justified.
Lines: 174-175.
Modification: “The Shapiro-Wilk test was carried out to check the normality of the sample distribution”.
- The manuscript lacks tables with basic data, anthropometric data as well as F-V test values ​​and MO data for the three groups of players under observation.
Lines: Between lines 102 and 103.
Modification: Expansion of Table 1 by introducing the data requested by the reviewer.
- At the end of the discussion, the strengths and limitations of the presented study should be mentioned in a separate paragraph.
Lines: 327 - 342.
Modification: Inclusion of the limitations and strengths section after the conclusions section
However, after discussion among members of the research team, we have decided not to follow the suggestion number five because we believe that the key correlation relationships are well explained throughout the manuscript.
Reviewer 2 Report
Comments and Suggestions for Authors
Dear authors,
It was a pleasure to review your manuscript in which relationships between maturity offset, anthropometric variables and the vertical force-velocity profile in youth male basketball players was examined. Overall, this manuscript is well-written and informative. I have some minor edits for the authors to help to improve the quality of the manuscript.
Abstract
Line 14- maximum theoretical force (F0), put the abbreviation in the first use above.
Introduction
Line 54- … decelerate better (thus protecting the leg structures)…” Expand on this? what do you mean by this? Did you mean preventing injuries?
Line 114- “profile determination coefficient, theoretical maximum power, force-velocity imbalance and type of de deficit”,…. Please correct this spelling error.
Results
I would suggest providing graphs to show the significant correlations as there are 3 crowded tables which makes it difficult for readers to pinpoint the significant outcomes.
Comments on the Quality of English LanguageQuality of English is acceptable.
Author Response
Response to manuscript reviewers:
Maturity offset, anthropometric characteristics and vertical force-velocity profile in youth basketball players.
Dear Editor,
Next, we describe the changes made to the manuscript following the recommendations made by the reviewers:
Reviewer 2
Dear review,
First, thank you very much for taking your time to help us to improve our manuscript. We have considered all your suggestions and contributions. Finally, we have accepted and agreed to modify the text by applying its first three recommendations.
- Line 14- maximum theoretical force (F0), put the abbreviation in the first use above.
Lines: 12 - 13.
Modification: “…(theoretical maximal force [F0], theoretical maximal velocity [V0] theoretical maximal power [Pmax], force-velocity imbalance [Fvimb], and force-velocity profile orientation)”.
- decelerate better (thus protecting the leg structures)…” Expand on this? what do you mean by this? Did you mean preventing injuries?.
Lines: 54.
Modification: “…more, accelerate more, decelerate with a better technique (reducing the risk of injury) or…”.
- “profile determination coefficient, theoretical maximum power, force-velocity imbalance and type of de deficit”,…. Please correct this spelling error.
Lines: 54.
Modification: “…more, accelerate more, decelerate with a better technique (reducing the risk of injury) or…”.
However, at your suggestion to provide graphs to show significant correlations because the tables contain a large amount of data, and this could confuse the reader (suggestion 5). The research group has decided not to follow this proposal. We appreciate your great contribution; we think it is very interesting but throughout the discussion we break down the significant correlations. We consider that including the graph would be explaining the same thing twice with a different instrument.
Acknowledgment
On behalf of the authors of this manuscript, I appreciate the advice and recommendations provided by the reviewers and the editor of the Journal of Functional Morphology and Kinesiology to improve the quality of the manuscript. It is a pride for this research team to be able to publish in your journal, and we will try to continue publishing in it.
Thank you.
Round 2
Reviewer 1 Report
Comments and Suggestions for Authors
The manuscript has been sufficiently improved to warrant publication in JFMK.